# Semi-visible jets, energy-based models, and self-supervision

Luigi Favaro[1], Michael Krämer[2], Tanmoy Modak[1], Tilman Plehn[1], and Jan Rüschkamp[1]

**1** Institut für Theoretische Physik, Universität Heidelberg, Germany
**2** Institute for Theoretical Particle Physics and Cosmology, RWTH Aachen, University, Germany

September 27, 2024

## Abstract

We present DarkCLR, a novel framework for detecting semi-visible jets at the LHC. DarkCLR uses a self-supervised contrastive-learning approach to create observables that are approximately invariant under relevant transformations. We use background-enhanced data to create a sensitive representation and evaluate the representations using a normalized autoencoder as a density estimator. Our results show a remarkable sensitivity for a wide range of semi-visible jets and are more robust than a supervised classifier trained on a specific signal.

# 1 Introduction

Model agnostic searches are of paramount importance for the current and future physics program at the Large Hadron Collider (LHC). The independence from signal hypothesis allows this approach to extend the coverage of possible new physics scenarios. Machine learning can provide a unique platform for this strategy by providing access to high-dimensional correlations and low-level data modeling.

The well-established approaches for model agnostic searches through anomaly detection are based on scores from density estimates or classification in semi-supervised settings between background and signal-enriched regions, see [1] for a recent review and [2] for an up-to-date list of relevant references.

Density-based scores select anomalies by identifying low-density regions of the data. Early research used the reconstruction error of an auto-encoder as a proxy for density [3,4]. In recent years [5–20], this approach has been continuously refined with better density estimates, such as in the normalized autoencoder (NAE) [21,22], normalizing flow techniques [17,23–25], energy flow polynomials [26], background estimation with ABCD methods [27], and network interpretability [28]. Anomaly detection through density estimation and semi-supervised learning has already been applied in recent ATLAS analyses [29,30]. More details on the different methods and architectures can be found in recent white papers [31,32].

However, the definition of an anomaly based on low-density regions of the data is not invariant under coordinate transformations [17,33]. Therefore, each step in the preprocessing chain can change what are considered inliers and outliers. To remedy this problem, we propose a framework for constructing a representation space suitable for anomaly detection in jet physics. We avoid the use of hand-crafted transformations of the data by creating observables based on physical invariances and a few assumptions about the signal hypothesis.

We develop our framework within a self-supervised contrastive learning representation (CLR) method CLR [34]. Self-supervision provides a unique way to detect anomalous objects in high-dimensional data. We generate "pseudo-labels" derived from the data, allowing the optimization of neural networks without relying on ground truth labels. This approach, similar to contrastive learning, can establish connections between original and augmented events, facilitating the discovery of novel phenomena. Learning invariances to transformation with contrastive learning has already been shown to be powerful in JetCLR [35], Anomaly-CLR [36], and resonant anomaly detection [37]. The latter introduces "anomalous" augmentations for anomaly detection applications on reconstructed high-level objects. These augmentations intentionally introduce variations in event kinematics that may resemble features found in anomalous events. Their definition follows general features of a new physics scenario and preserves the model agnostic aspect of an unsupervised anomaly detection tool.

In this work, we apply the concept of anomalous enhancements to the detection of semi-visible jets [38–44]. Semi-visible jets arise in models of strongly interacting dark sectors, which in turn belong to the general class of Hidden Valley models [45–47]. Distinguishing such semi-visible jets is difficult and represents a major challenge for jet classification. The main background is the production of light quark jets due to Quantum Chromodynamics (QCD) effects, also referred to QCD jets. We call our framework DarkCLR, an extended representation space for studying and finding semi-visible jets within LHC jets. We show that the latent space learned by DarkCLR provides informative representations of semivisible jets for downstream tasks. We propose two scores for anomaly detection: an anomaly score defined in the representation space, and the reconstruction error of a normalized autoencoder trained on the representations.

Our paper is organized as follows. We describe the background data and signal benchmarks in Sec. 2. Then, Sec. 3 introduces DarkCLR, the network architecture, and the physical and

anomalous extensions. We present the anomaly scores in Sec. 4, and finally, we look at the tagging performance in Sec. 5, examining the discriminative power of the representations, the robustness of the anomaly scores, and the dependence on the main training hyperparameters.

## 2 Dark jets

Jets are a prevalent signature of several new physics models, such as Hidden Valley models, which can lead to tantalizing semi-visible jet signatures at the LHC. In this work, we are interested in Hidden Valley models that consist of a strongly coupled dark sector with dark quarks coupled to the Standard Model (SM) through a vector mediator. As a result, jets can be produced by the dark quarks from the decay of the vector mediator. The shower in this case would involve radiation into the dark sector, resulting in jets that are called semi-visible or dark jets, depending on the phenomenology of the signal.

For our purposes, we consider a benchmark signal scenario with an underlying dark sector as introduced in [15, 17, 43]:

$$pp \rightarrow Z' \rightarrow q_d \bar{q}_d, \;\; \text{with} \;\; m_{Z'} = 2\,\text{TeV} \;\; \text{and} \;\; q_d = 500\,\text{MeV}, \tag{1}$$

where $Z'$ is the mediator between the dark sector and the SM quarks, charged under a $U(1)'$ gauge group, and $q_d$ is a dark quark charged under a dark $SU(3)_d$. The dark sector hadronizes to dark pions ($\pi_d = 4$ GeV) and dark rho mesons ($\rho_d = 5$ GeV). The neutral dark rho mesons mix with the $Z'$ and can thus decay into SM quarks. The other dark mesons are stable and escape detection. In our benchmark scenario the fraction of invisible particles in a shower is given by $r_{\text{inv}} = 0.75$ [15, 43]. This dark sector model then leads to semi-visible jets and can be simulated with the Pythia Hidden Valley module [48, 49]. We will refer to this benchmark scenario as the "Aachen" dataset in the remainder of the paper.

The dataset is generated using Madgraph5 [50] for the hard process. The generated events are then interfaced with Pythia 8.2 [51] for showering and hadronization and finally fed to Delphes 3 for fast detector simulation [52]. The jets are reconstructed using the anti-$k_T$ algorithm [53] with radius parameter $R = 0.8$ in FastJet [54].

The most important phenomenological parameters for Hidden Valley models are the invisible fraction of the constituents, $r_{\text{inv}}$, and the mass of the dark mesons, $m_{\pi/\rho}$. To test the model dependence of our approach, we generate several data sets with the following parameter choices: starting from our benchmark signal, we first vary only the mass of the dark mesons and the confinement scale $\Lambda$ as $m_{\pi_d} = m_{\rho_d} = \Lambda = 10\,\text{GeV}, 20\,\text{GeV}$. In addition, for our default choice of dark meson masses, we change the invisible fraction $r_{\text{inv}}$ by allowing all dark mesons to decay back to SM quarks with a given probability. To explore the region where the number of visible jet constituents is closer to the QCD background, we reduce the invisible fraction to $r_{\text{inv}} = 0.5, 0.2$. The light QCD background is generated from leading order di-jet events.

The selection of the jets at detector level is done by calculating the $\Delta R$ between the reconstructed fat jets and the dark quarks at parton level and ensuring that $\Delta R < 0.8$. On the selected fat jets we apply a kinematic selection in $p_T$ and $\eta$, namely

$$p_T^j = 150...300 \text{ GeV} \;\; \text{and} \;\; |\eta^j| < 2. \tag{2}$$

# 3 DarkCLR

## 3.1 Contrastive Learning Representation

Contrastive Learning of Representations (CLR) is a method for learning representations of the training data in high-dimensional spaces. These representations can then be used for any downstream task, from classification to unsupervised learning. CLR falls into the category of self-supervised learning, i.e. it does not require "truth" labels of the training data.

In CLR, a function $f(\cdot)$ maps from the data space $\mathcal{D}$ to a representation space $\mathcal{R}$, where the function is optimized to solve an auxiliary task for which we define pseudo-labels. In this work, we focus on performing anomaly detection on the representations. Therefore, the function that performs the mapping from $\mathcal{D}$ to $\mathcal{R}$ is trained only on background data. Since collider events or objects such as jets typically consist of unordered sets of particles, we opt for a permutation invariant architecture. Specifically, we use a transformer encoder network to learn the mapping.

To overcome the lack of signal in our training data and to keep the approach model agnostic, we use only augmentations of the background data. These augmentations are used to define two types of pseudo-labels:

- **Positive-pair:** $x_i, x_i'$. This pair is constructed from a data point and an augmented version of itself via a positive augmentation;
- **Anomaly-pair:** $x_i, x_i^*$. This pair is constructed from a data point and an augmented version of itself via an anomalous augmentations.

Once we have defined the pseudo-labels, we minimize the following loss function [36]:

$$\mathcal{L}_{\text{AnomCLR}}^+ = -\log \ e^{\left(s(z_i, z_i') - s(z_i, z_i*)\right)} = s(z_i, z_i^*) - s(z_i, z_i'), \tag{3}$$

where $z_i = f(x_i)$, $z_i' = f(x_i')$, $z_i^* = f(x_i^*)$ and $s(\cdot, \cdot)$ is the cosine similarity, a measure of proximity between points in a compact $\mathbb{S}^{d-1}$ representation space. The function $f(\cdot)$ then maps the raw data into the representation space such that positive pairs are close in $\mathcal{R}$ while anomaluous pairs are pushed apart. The first objective is commonly known as alignment and the latter one ensures separation between objects in the anomalous pair. While the term $s(z_i, z_i')$ ensures the alignment, i.e. different objects are mapped onto the same point in the compact latent space, the term $s(z_i, z_i^*)$ maximizes the distance between anomalous pairs while keeping the representation space informative about the anomalous augmentations. The chosen transformations are intended to be alterations of the original data that preserve the fundamental physics, such as the symmetries of the system. More details about the applied augmentations are given in the next section.

Note that $\mathcal{L}_{\text{AnomCLR}}^+$ is a modified version of the original CLR loss function [36] and has two special features that we can exploit. First, it contains only the invariances we want to impose and the anomalous features we want to distinguish from the background. Therefore, the representation space will be approximately invariant to the symmetries of the data we require during training, and it will be exposed to potential new physics signals through the anomalous augmentations. Second, as shown in Eq. (3), the loss function scales as $N_{\text{batch}}$, as opposed to the $N_{\text{batch}}^2$ scaling of the original CLR loss function [36], and is therefore less computationally expensive. Although the partial removal of the uniformity requirement could potentially lead to a collapse of the representation space to a single point, this is not observed in our numerical analysis. We suggest that the large variety in the training data combined with the use of multiple augmentations prevents mode collapse and information loss.

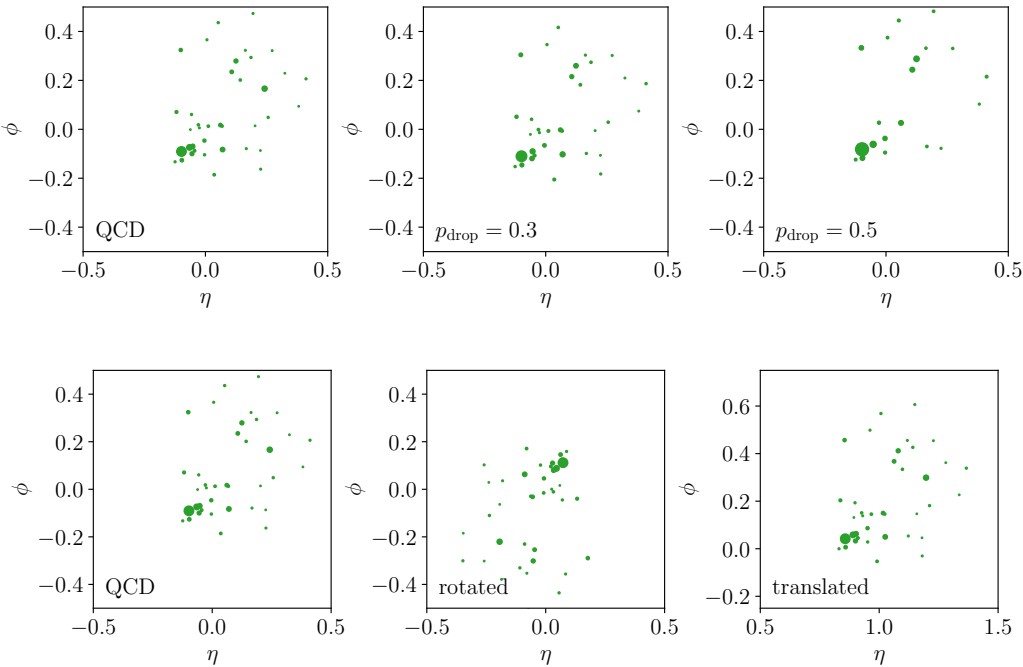

Figure 1: (top) Example of positive augmentations on a QCD jet. The original QCD jet is rotated in $\eta - \phi$ in the middle panel and translated in $\eta - \phi$ in the right panel. (bottom) Example of an anomalous transformation on a QCD jet. The left panel shows the original background jet, while the middle and right panels show the same jet after applying the augmentations with $p_{\text{drop}} = 0.3$ and $p_{\text{drop}} = 0.5$ respectively.

## 3.2 Augmentations

Here we discuss the augmentations we use during training. We start with the positive (or, synonymously, physical) augmentations. These are easy to implement approximate symmetries of a jet:

- **Rotations:** We rotate each jet in $\eta - \phi$ by an angle which is chosen randomly between $[0, 2\pi]$. Note that the angle is chosen randomly for each jet, i.e., each constituent inside a jet is rotated by the same angle.

- **Translations:** We shift each constituent in the $\eta - \phi$ plane by randomly choosing a shift in a window with size given by the distance between the two furthest constituents.

After applying these two transformations to the original jet $x_i$, we obtain the augmented version $x_i'$ and the positive pair $\{x_i, x_i'\}$.

Semi-visible jets, as discussed earlier, have fewer constituents than QCD jets. Therefore, we consider the dropping of constituents as a **anomaly augmentation**. The transformation is implemented as follows: We drop each component of the jet with a fixed probability $p_{\text{drop}}$, and the $p_T$ of the augmented jet is rescaled to match the original $p_T$. The latter step ensures that the augmented jets fulfill the selection cuts applied in the generation process.

Fig. 1 shows an example transformation of a QCD jet used during training with $p_{\text{drop}} = 0.3$ and $p_{\text{drop}} = 0.5$.

## 3.3 Network architecture

We describe the network architecture following the schematic in Fig. 2. Note that in this section the indexed variable $x$ refers to the constituents of a single jet and not to one instance of the

Figure 2: Schematic of the network architecture. The shape of the input vector, excluding the batch dimension, is shown after each step.

training data.

As the first step of the CLR network, an embedding layer maps the set of constituents $\{(p_{Ti}, \eta_i, \phi_i)\}_{i=1}^{N_c}$ to a larger vector with $d_r = 128$ dimensions. The number of selected constituents has a fixed maximum size of $N_c = 50$. This selection will in most cases include the entirety of the QCD jets, while ignoring the softest components if the number of constituents is larger than $N_c$. The embedded constituents are then passed through a sequence of transformer encoder blocks. A block consists of a multi-head self-attention layer followed by a feed-forward network. A single-head self-attention operation transforms the input set by taking into account all correlations between the constituents. It is mathematically expressed as

$$
\begin{aligned}
x_i' &= \sum_{j=1}^{N_c} a_j v_j \\
&= \sum_{j=1}^{N_c} \text{Softmax}_j \left( \frac{(W^Q x_i) \cdot (W^K x_j)}{\sqrt{d_z}} \right) W^V x_j ,
\end{aligned}
\tag{4}
$$

where $W^Q$, $W^K$, and $W^V$ are learnable matrices, and $d_r$ is a normalization factor equal to the dimensionality of the query $x_i$. The Softmax operation ensures that the $a_j$ are a set of weights, also called attention weights, which are applied to the vector $v_j$. The multi-head operation simply splits the self-attention into separate learnable weight matrices over the embedding/output dimension. The output of the last transformer block provides an encoding of dimension $(N_c, d_z)$. As a crucial next step, the output is summed over $N_c$ to induce permutation symmetry between the constituents. Finally, the output is passed to a fully connected head network. The output of the head network then serves as the representation and input to the contrastive loss function of Eq. 3. Unless otherwise stated, the set of parameters used to train the transformer network is summarized in Tab. 1.

If a jet has less than $N_c$ constituents, these are zero-padded. We ensure that this does not affect the transformer by masking the zero $p_T$ entries. The masking procedure ensures that the attention weights from zero-padded constituents are zeros by adding minus infinity to the attention weight before normalization. Additionally, the contributions from the masked particles are also ignored in the final aggregation over $N_c$ [35].

## 4 Anomaly scores

**CLR anomaly score** We study the effect of the CLR transformation by analyzing the CLR embedding space. We show in App. A that the representation before the head network encodes useful information for the discrimination between background and signal. In particular, the representations perform better than the constituent-level on a simple linear classifier test (LCT). However, we find that the output of the head network performs better on a cut-based analysis on a very simple quantity, and we use this representation for the evaluation of the anomaly scores. We first note that one way to reduce the loss is to simply increase the length of the vector so that jets with different properties are separated in the non-normalized space and close to each other after projection. Therefore, we expect the norm of the representation

| Hyper-parameter | Value |
| --- | --- |
| Embedding dimension ($d_r$) | 128 |
| Feed-forward hidden dimension ($d_z$) | 512 |
| Output dimension ($d_z$) | 512 |
| # self-attention heads | 4 |
| # transformer layers ($N$) | 4 |
| # head architecture layers | 2 |
| Dropout rate | 0.1 |
| Optimizer | Adam ($\beta_1 = 0.9$, $\beta_2 = 0.999$) |
| Learning rate | $5 \times 10^{-5}$ |
| Batch size | 256 |
| # constituents ($N_c$) | 50 |
| # jets | 100k |
| # epochs | 150 |

Table 1: Default configuration of the transformer encoder and the training process.

vector to be a discriminative scalar quantity and propose it as a CLR-based anomaly score that can show the effect of the DarkCLR pretraining. Namely:

$$s_{\text{CLR}} = ||z||_{L_2}, \qquad z \in \mathbb{R}^{d_z}. \tag{5}$$

Before using this anomaly score, a small modification is needed. Since our loss Eq. 3 is norm-free, the ordering between background and signal norms is not a priori fixed. This ambiguity, which can spoil applications in anomaly detection, is resolved by introducing a regularization term that penalizes background representations with large norms. This ensures that anomaly detection associates high norm with outlier data. The implementation is done by adding the $L_2$ norm of the representations of the background batch to the loss function. We find empirically that this new term does not affect the similarity, and therefore the loss, of the training. By definition $s_{\text{CLR}}$ has no access to angular information which should provide additional discriminative information. We include this in the following anomaly score, which takes the full high-dimensional vector as input.

**NAE**    The second anomaly score we consider is the reconstruction error of an autoencoder. In an autoencoder we define an unsupervised learning task by constructing an encoder and a decoder network trained only on the background data. The compression that takes place in the encoder forces the network to learn the manifold of the dataset in a latent space from which the decoder has to reconstruct the original input. This is achieved by minimizing the reconstruction error of the input, where we follow the standard practice of using the mean squared error as a measure of the reconstruction quality. After training, we can use the same quantity as an anomaly score, since off-manifold events are not reconstructed by the decoder, and thus give a large reconstruction error.

In particular, we use a statistically well motivated version of an autoencoder, the normalized autoencoder (NAE) [21, 22]. A normalized autoencoder promotes classical autoencoder training to an energy-based model by fixing the energy function to be the reconstruction error of the network. An NAE has the same structure as a standard AE with the added robustness of Maximum Likelihood Estimate (MLE) training. The strategy for the analysis of the Dark-CLR representations consists of obtaining the latent representations via the encoding function $f$ defined in Sec. 3 and pass them to the NAE. The energy function is then used as anomaly score, which is approximately invariant under the physical transformations used during the

CLR training. Since the autoencoder is trained in a second step, DarkCLR can be seen as a pre-training procedure which exploits known invariants and the anomalous augmentation to provide better representations when we run a density estimation downstream task.

In the following description of the NAE methodology, we assume to train on representations $z$ sampled from the latent space distribution $p_Z(z)$ induced by the training data,

$$z = f(x) \qquad \text{where} \qquad x \sim p_{\text{data}}(x). \tag{6}$$

In the NAE we assume a Boltzmann underlying distribution $p_\theta$ with energy $E_\theta$:

$$p_\theta(z) = \frac{e^{-E_\theta(z)}}{\Omega}, \qquad E_\theta(z) = ||z - z'||_2, \tag{7}$$

where $\theta$ are the trainable parameters of the network, and $z'$ is the reconstructed representation.

Performing MLE on the probability distribution translates to minimizing the sum of the reconstruction error and the normalization factor $\Omega$. However, computing $\Omega$ becomes easily intractable for high-dimensional spaces, so we do not explicitly minimize this quantity. Instead, we rewrite the gradient of a maximum likelihood loss function in a computationally feasible manner as [21]:

$$\begin{aligned}
\nabla_\theta \mathcal{L} &= \mathbb{E}_{z \sim p_Z}[-\nabla_\theta \log p_\theta(z)] \\
&= \mathbb{E}_{z \sim p_Z}[\nabla_\theta E_\theta(z)] - \mathbb{E}_{z \sim p_\theta}[\nabla_\theta E_\theta(z)].
\end{aligned} \tag{8}$$

This allows us to reformulate the optimization as a min-max problem, where samples from the model distribution substitute the expensive integral. We obtain samples from $p_\theta$ using Langevin Markov Chains (LMC). An LMC process follows the equation:

$$z_{t+1} = z_t - \lambda \nabla_z \log p_\theta(z) + \sigma \epsilon \qquad \epsilon \sim \mathcal{N}(0,1), \tag{9}$$

and does not require an estimate of the integral due to the independence of the latter from the input $z$.

In particular, we utilize the Contrastive Divergence (CD) [55] Markov chain Monte Carlo scheme. Given a transition kernel $T_\theta$ for the data distribution $p_Z$, the following combination of Kullback-Leibler (KL) divergences has a zero only for $p_\theta(z) = p_Z(z)$ [56]:

$$D_{\text{KL}}(p_Z || p_\theta) - D_{\text{KL}}(T_\theta^t(p_Z) || p_\theta), \tag{10}$$

Therefore, we can run short Langevin Markov Chains with steps $t$, which define the transition kernel $T_\theta^t$, and estimate the gradients of Eq. 8 as:

$$\nabla_\theta \mathcal{L} = \mathbb{E}_{z \sim p_Z}[\nabla_\theta E_\theta(z)] - \mathbb{E}_{z \sim T_\theta^t p_Z}[\nabla_\theta E_\theta(z)]. \tag{11}$$

Note that Eq. 11 ignores an additional term as pointed out in [55]. We find that this approximation does not affect the convergence of our model and therefore we use the base CD loss.

The procedure defined above stabilizes the training and corrects for the mismodeling of the density estimate introduced by the mere minimization of the reconstruction error. The epoch with the energy difference closest to zero defines the best loss, and we select the corresponding model for evaluation. Before turning on the regularization term, we pre-train the autoencoder for 200 epochs then continue training according to Eq. 8 for another 100 epochs. The architecture of the encoder network is a simple feed-forward network with five layers with neurons from 128 to 8 in powers of two and a three-dimensional bottleneck. The decoder mimics the encoder network, this time up-sampling from 8 to 128 dimensions in powers of two.

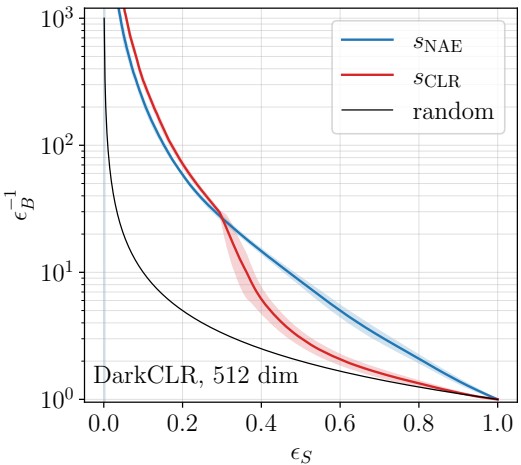

Figure 3: ROC curves of background suppression $\epsilon_B^{-1}$ versus signal efficiency $\epsilon_S$, computed from the $L_2$ norm of the representations, $s_{\text{CLR}}$ (red), and from the MSE of the NAE trained on the representations from DarkCLR, $s_{\text{NAE}}$ (blue).

|  | DVAE [28] | INN [17] | NAE Jet images [21] | DarkCLR |
|---|---|---|---|---|
| AUC | 0.71 | 0.73 | 0.76(1) | 0.76(1) |
| $\epsilon_B^{-1}(\epsilon_S = 0.2)$ | 36 | 39 | 41(1) | 59(1) |

Table 2: Summary of AUCs and background rejections at low signal efficiencies for DarkCLR compared to other methods. The numbers in parentheses indicate the standard deviation of the score from an ensemble of networks. For the DVAE and the INN this was not reported.

## 5 Results

In this section, we show results using DarkCLR on the benchmark signal. First, we compare our results with previous methods tested on the same dataset. We then perform studies to test the robustness of our results with respect to variation of the semi-visible jet model parameters. Finally, we discuss the dependence of the performance on the main network parameters.

### 5.1 Improved performance

First, we discuss the base pipeline of our procedure and compare the results with other methods. We train the transformer encoder network with the hyper-parameters as specified in Tab. 1. The chosen embedding space uses 512 dimensions, and the augmentations follow the implementation described in Sec. 3, where $p_{\text{drop}} = 0.5$. Note that the size of the embedding space must be large enough to contain the information passed from the head to the output layer. As we show in App. A, our results are not sensitive to the specific choice of the embedding dimension, as long as it is sufficiently large. We show Receiver Operator Characteristic (ROC) curves for the CLR latent score $s_{\text{CLR}}$ and the NAE score $s_{\text{NAE}}$. In addition, we report the low signal efficiency background rejection as a measure of the purity of a signal sample in the low background region and the area under the curve (AUC) score. The error bands on $s_{\text{CLR}}$ are taken from 5 runs of CLR training with different initializations. From each of these representations, we train 3 autoencoders for a total of 15 $s_{\text{NAE}}$ scores, which are used to compute the

mean and standard deviation. Note that no transformations are applied to the representations before training the autoencoder, thus limiting the preprocessing to the mere $p_T$ rescaling and the physically guided CLR transformation.

Fig. 3 shows the ROC curves obtained with our method. The new embedding space greatly improves the background rejection $\epsilon_B^{-1}$, in particular in the region of low signal efficiency as estimated by $\epsilon_B^{-1}(\epsilon_S = 0.2)$. We find that the transformer network does indeed encode information in the norm to discriminate between jets. In particular, it improves purity in the low background region, as shown by the background rejection of $s_{\text{CLR}}$ at low signal efficiency. However, due to the high dimensionality of the representations, many jets will share the same norm in the bulk of the distribution, causing the $s_{\text{CLR}}$ ROC curve to drop off at $\epsilon_S = 0.3$. We also observed similar problems when training a standard autoencoder. This is solved by a more precise density estimator like the NAE. The resulting $s_{\text{NAE}}$ ROC curve is much more stable with an average AUC of 0.76 and a $\epsilon_B^{-1}(\epsilon_S = 0.2) = 59$.

Tab. 2 summarizes the AUC and the background rejection $\epsilon_B^{-1}(\epsilon_S = 0.2)$ for DarkCLR and compares them to previous methods: an NAE trained on jet images [21], a Dirichlet variational autoencoder [28], and an invertible neural network [17]. While the best AUC is similar for all methods, with DarkCLR we find much stronger background rejection at low signal efficiency, and we do not rely on image-based representations or any specific preprocessing steps.

## 5.2 Robustness of DarkCLR

**Dependence on the dark shower signal** As a next step, we study the robustness of our method with respect to the main phenomenological parameters of the semi-visible jet as described in Sec. 2. We set up a benchmark by training a transformer classifier with 100k jets equally divided between the QCD background and the "Aachen" dataset. We then use the classifier score to detect the signals with different invisible fraction $r_{\text{inv}}$ and dark meson mass scale $m_{\text{mesons}}$. The classifier uses the same backbone transformer architecture of Sec. 3 where the head network is replaced by a two-layer MLP with ReLU nonlinearities and a single output. We train the network for 300 epochs, minimizing the binay cross-entropy loss, and refer to the validation loss to select the best model.

Fig. 4 shows the results of the supervised classifier (left panel) compared to DarkCLR trained only on the QCD background and tested on all signals (right panel). The supervised classifier shows a large drop in performance when applied to datasets with different model parameters, see also [15]. Instead, our DarkCLR method performs well on different semi-visible jet signals, as expected from the unsupervised training approach.

The small differences between the DarkCLR ROC curves for the various signals can be understood by analyzing the phenomenological aspects of the different semi-visible jet models. As we reduce the invisible fraction $r_{\text{inv}}$, the signal becomes more similar to a QCD jet, increasing the overlap between the two distributions and thus reducing the detection efficiency. Similarly, increasing the confinement scale and thus the mass of the dark hadrons leads to an earlier hadronization of the dark quarks. Therefore, the visible SM decays continue to shower down to the QCD confinement scale, again more closely resembling a QCD background jet initiated by light quarks. We observe this effect when we increase the energy scale from the default choice of the Aachen benchmark dataset to $m_{\pi_d} = m_{\rho_d} = \Lambda = 10\,\text{GeV}$ and $20\,\text{GeV}$.

For a summary of the background suppression at low signal efficiency, see Tab. 3. The generalization capabilities of DarkCLR outperform the supervised classifier for all signal models, especially in the more interesting low signal efficiency region.

**Impact of Anomaly augmentation** To validate the use of anomalous augmentations, we compare DarkCLR with the standard JetCLR training. The latter is trained only on QCD jets using the set of physical augmentations. We refer to previous work for the implementation

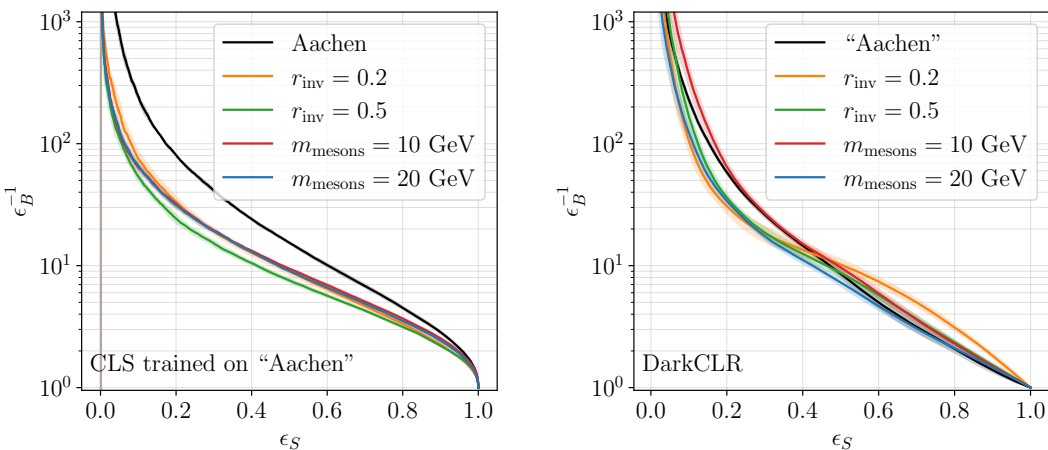

Figure 4: Left panel: ROC curves of a supervised classifier trained on the "Aachen" benchmark signal and tested on datasets with different dark shower model parameters. Right panel: ROC curves obtained from DarkCLR after training on the QCD background only and tested on additional datasets.

| | $\epsilon_B^{-1}(\epsilon_S = 0.2)$ | | | | |
|---|---|---|---|---|---|
| | "Aachen" | $r_{\mathrm{inv}} = 0.2$ | $r_{\mathrm{inv}} = 0.5$ | $m_{\mathrm{mesons}} = 10$ GeV | $m_{\mathrm{mesons}} = 20$ GeV |
| CLS ("Aachen") | 80(1) | 28(2) | 22(2) | 30(2) | 28(2) |
| DarkCLR | 58(2) | 28(2) | 35(3) | 65(7) | 33(1) |

Table 3: Summary of the results presented in Fig. 4 for the background rejection $\epsilon_B^{-1}$ at a signal efficiency of $\epsilon_S = 0.1$.

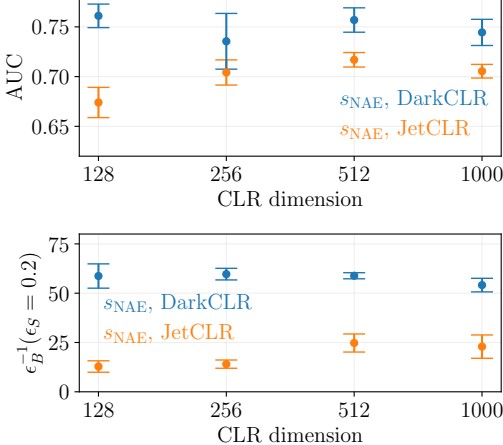

Figure 5: CLR and NAE AUC (upper panel) and background rejection at low signal efficiency (lower panel) for different embedding dimensions.

and training of JetCLR [35]. After creating the new representations, we train an NAE using the same procedure. Fig. 5 shows the performance of JetCLR compared to DarkCLR in terms of AUC and background rejection for the benchmark dataset. Without anomalous pairs, the results vary between different embeddings and underperform in both figures of merit. Notably, DarkCLR improves detection at low signal efficiency even for small embedding dimensions, while without augmentation we observe a small increase in sensitivity only for large embedding spaces.

# 6 Summary and outlook

In this article we present DarkCLR[§], a new framework for detecting semivisible jets at the LHC, as predicted in models with a strongly interacting dark sector. DarkCLR is a self-supervised method based on contrastive learning representations. The CLR paradigm provides a new representation that is approximately invariant under physically motivated transformations of the data. In this study, a permutation invariant network learns a jet representation that is invariant to rotations and translations in the angular coordinates.

In general, preprocessing can improve the discrimination between QCD background and dark shower signals. However, the preprocessing is often hand-crafted and model-specific, and the performance of the classifier depends on the chosen transformations. We propose to introduce an augmented anomalous feature in the CLR training to learn such preprocessing based on general physical features of the signal. For semivisible jets, this is done by introducing an anomalous augmentation that drops components from the original jet. This ensures that the training uses only background events, reducing the dependence on the details of the dark sector model.

We show that the transformer network provides a discriminative representation of the data, which we use for unsupervised anomaly detection with a normalized autoencoder. Our method does not rely on hand-crafted preprocessing or an image representation of jets, and exhibits stronger background rejection at low signal efficiency compared to previous state-of-the-art methods. The probability distribution of the representations is not modified before training the autoencoder, thus limiting the effect of coordinate transformation on physically motivated CLR training.

In addition, we test the dependence of our model on the main phenomenological parameters entering the dark shower model, the invisible fraction of particles and the mass of dark mesons. We find that a supervised classifier is highly sensitive to the specific choice of signal parameters used during training, especially at low signal efficiencies. In contrast, our method, based on a density estimation of the background, is more robust to a variation of the parameters of the dark shower model, thus validating the application of unsupervised methods for a model agnostic search. In our experiments we assumed uncorrelated visible constituents, i.e. constituents are uniformly dropped in the jet. A dedicated study on this specific signature is needed to evaluate any potential bias. However, our framework is flexible enough to account for this modifications. Both positive and anomalous augmentations can be extended to cover different transformations, e.g. detector smearing effects or non-uniform decay of dark sector particles back to the SM.

We provide a proof-of-concept application of self-supervision for the detection of semivisible jets. Further studies will include the inclusion of additional augmentations for a wider coverage of signal classes where jet multiplicity is not the leading discriminative feature. We will also investigate the effect of choosing the dimensionality of the representation space and the interpretability of the latent space. More generally, although we have based our studies on

---

[§]The code and the data are available at https://github.com/luigifvr/dark-clr

simulations, we foresee the application of DarkCLR directly on data to overcome the effects of particular simulation choices, e.g. a specific hadronization model for the dark sector.

## Acknowledgements

We would like to thank Barry Dillon for many useful discussions. LF would like to thank Alexander Mück and Elias Bernreuther for helping with the generation of the dark showers. We would like to thank the Baden-Württemberg-Stiftung for financing through the program *Internationale Spitzenforschung*, project *Uncertainties – Teaching AI its Limits* (BWST_IF2020-010). This research is supported by the Deutsche Forschungsgemeinschaft (DFG, German Research Foundation) under grant 396021762 – TRR 257: *Particle Physics Phenomenology after the Higgs Discovery*.

# A   Linear classifier test (LCT)

As a final study of the separability between QCD and semi-visible jets, we train a Linear Classifier Test between background and signal. Even though we move to a supervised scenario, the network never accesses the signal data during training. This evaluation will test the separation power and the information content in the representations starting only from QCD jets and their augmentations. We disentangle the effects of the embedding dimension and the head network by selecting 128 as the embedding dimension of the transformer and scanning over the output dimension of the head network. This choice closely matches the original dimensionality of the input data. Fig. 6 (left) shows that the LCT of the head representation is informative regardless of the output dimension. The head network is affected by the projection on the hypersphere and requires a larger dimension to saturate to the same separation power. In both cases, we observe that the representation space is simpler than the original constituent-level space. The implemented LCT is a single linear layer network without non-linearities.

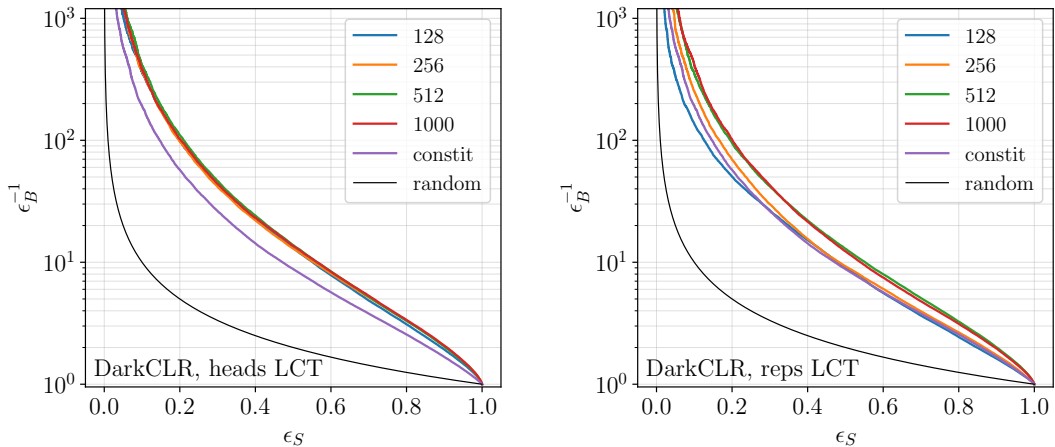

Figure 6: Linear Classifier test between the Aachen benchmark dataset and QCD jets. Head representations (left) and output representations (right) with different embedding dimensions from 128 up to 1000. The LCT on raw constituents is shown in purple.

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
