# Peer review of "Semi-visible jets, energy-based models, and self-supervision"

_SciPost Physics_

## Round 3 · Referee Report · Anonymous (Referee 1) · 2024-10-24

Strengths

The paper describes a new tagging algorithm for identifying semi-visible jets
based on a contrastive learning representation.
1 - The algorithm presented is interesting and relevant, relying on minimal physical features of the signal.
2 - The background rejection is superior to supervised classifiers
and more stable with respect to changes in the model parameters.
3 - The text is well written and the results presented strongly support the
paper claims.
4 - The paper is clear and accessible to non-experts on current machine learning developments.

Report

The paper clarity has been improved, making it accessible to non-experts on the latest machine learning developments.

Requested changes

The top/bottom plots in figure 1 are switched with respect to their descriptions in the caption. Also, the last sentence of Sec.3.2 should read "Fig.1 (bottom) ....".

Recommendation

Publish (surpasses expectations and criteria for this Journal; among top 10%)

---

## Round 3 · Author Response

We would like to thank the referees for their detailed comments and feedback.

In response to the main points raised by the referees, we have significantly improved the clarity of the paper. The main changes are a new schematic diagram of the network and a discussion of the transformations in each block. We have also added a summary of the strategy for the NAE in Section 4 specifying that the anomaly score is now approximately invariant under the positive transformations.

We found it difficult to study the bias of non-uniform decay to the Standard Model without a specific benchmark signal. We tested the trained models on mock datasets in which we dropped a large fraction of the constituents from a single reclustered sub-prong of the QCD dataset, and found that the network was sensitive to this. However, this test uses data already seen during training and should be complemented with a case signal model. We note that this is not a limitation of our framework. DarkCLR allows the implementation of additional augmentations, positive or anomalous, covering other modes in the missing energy distribution within the jet. For example, Ref.[36] used four anomalous augmentations and retained robustness over a wide range of signals. We have added a comment in the conclusions.

To improve the reproducibility and future developments, we published the dataset on Zenodo, DOI: 10.5281/zenodo.12801842.

---

## Round 3 · List of Changes

Report 1
1. See main reply

2. We observed with the linear classifier test that the representations before and after the head network have similar performance. However, we think that the behaviour of the final representation is easier to understand because it is directly related to the CLR loss, i.e. the learning invariants. We also observe that a cut-based analysis on the norm of the vector, a very simple scalar function, can detect outliers. We expect the vector information to improve the sensitivity. For example, this information is included in the autoencoder, which works on the full vector. We have modified the sentence to better reflect this idea and added a reference to the linear classifier test appendix.

3.
We have clarified this issue in the summary of the NAE in section 4.

4.
We have changed the labels to "DarkCLR" and "JetCLR". This is now consistent with the text, and we use the same signal efficiency threshold throughout the paper.

5.
We have corrected this typo.

6.
The numbers in parentheses are the one sigma deviations from the ensemble on trained networks. The other entries in the table are copied from previous papers where an error was not reported. This is now explicit in the text, and we have included error bars
on the classifier in Section 5.2

7.
We have removed the label for section 2.1

8.
This is indeed a reconstruction cut. We have moved the discussion of the cut to after the discussion of the clustering details.

9.
The rescaling is done to ensure that the selection cuts in the augmented dataset are the same as in the original data. If this were not the case, the network would have an easy variable to discriminate between the two samples, which is not consistent with the construction of powerful observables. We have added a comment to the description of the augmentation.

10.
This is a general notation for: "distributed according to".

11.
We have defined this acronym and a few others the first time they are used in the text.

Report 2

1.
We have included a figure showing the positive augmentations in Figure 1.

2.
See main reply

3&4.
See main reply

5.
We have defined this acronym and a few others the first time they are used in the text.

---

## Editorial Decision

resubmitted